# OpenReview forum: "TAI3: Testing Agent Integrity in Interpreting User Intent"
_NeurIPS.cc/2025/Conference — NeurIPS 2025 poster_

### Official Review · Reviewer_HfFD · 2025-07-01

**Clarity:** 4
**Significance:** 4
**Originality:** 4
**Rating:** 5
**Confidence:** 4

**Summary:**

TAI3 is a framework for stress-testing LLM agents in intent understanding. TAI3 employs semantic partitioning and intent-preserving mutations to create natural language inputs that the LLM agent may be prone to misinterpreting. The authors show that TAI3 outperforms self-reflection approaches in identifying intent-interpretation errors, and identifies more meaningful (broad-coverage) errors more efficiently (fewer queries).

**Questions:**

* When is $\mathcal{I}$ used to describe user intent and when is it used to describe intent category? As mentioned earlier, L211 and L220-223 are examples of lines where $\mathcal{I}$ seems to describe user intent, not intent category-- but L181 says that $\mathcal{I}=\{VA,IV,US\}$.

* What are the error margins on the numbers in Fig 6?

* How does TAI3 scale when the parameter x category x intent table becomes quite large? Is there a mechanism to prioritize certain cells, or possibly test multiple cells with the same input?

**Ethical Concerns:**

["NO or VERY MINOR ethics concerns only"]

**Final Justification:**

After clarifications and added details and reviewing the authors' conversations with other reviewers, I maintain my score.

**Limitations:**

yes

**Quality:**

4

**Strengths And Weaknesses:**

## Well-scoped topic, appropriate experimental setup

The paper identifies an important topic, clearly defines the scope of the project, and presents a clean problem formulation to address it. The research questions are clearly stated and cover the questions of interest for this topic. The experiments then answer the questions well and present the results well.

## Simple, performant method

The method is simple, requires little human involvement, and is shown to outperform the presented baselines. Most deltas are large enough to not need confidence tests, but the numbers in Figure 6 are quite close. Including statistical significance tests / error margins in general would be appreciated.

## Inconsistent use of "intent"

The term "intent" is used heavily in this paper. Sometimes "intent" means user intent as in what the user would like done by the agent. Sometimes "intent" is the categories of valid, invalid, and underspecified. For example, L211 says that semantics-preserving manipulations preserve $\mathcal{I}(u)$, which according to L181 is the intent categories.

---

> ### Author Rebuttal · Authors · 2025-07-31
>
> We greatly appreciate the reviewer’s supportive and insightful feedback. We will incorporate all suggested changes during revision.
>
> ---
>
> **Q1: Inconsistent use of "intent"**
>
> **A1**:  Thanks for pointing this out. The notation $\\mathcal{I}$ should be intent category, as defined in L181.
>
> To avoid confusion, in L218-221, we will add one new notation $\\mathcal{T}(u)$ to denote user intent, namely the tokenized representation of user task $u$.
>
> Accordingly, we will revise Equation. 1 as follows:
>
> $\\sum\_{i}^{|\\mathcal{T}(u)|} \\log P(\\mathcal{T}(u)\_i|\\mathcal{T}(u') \\cdot \\mathcal{T}(u)\_{ \< i}; \\theta)$
>
> Where $u$ denotes seed task and $u’$ is the mutated task. The operator $\\cdot$ denotes sequence concatenation.
>
> We will update the notations for better clarity during revision.
>
>
> ---
>
>
> **Q2: Error margins on the numbers in Fig 6?**
>
> **A2**: We show the error margins of Figure 6 in the table below. We will update the figure during revision.
>
> |      AQFF $\downarrow$                |           VALID           |          INVALID          |         UNDERSPEC         |
> | :-------------- | :----------------------- | :----------------------- | :----------------------- |
> |    **Model**     |    **SelfRef / Ours**     |    **SelfRef / Ours**     |    **SelfRef / Ours**     |
> | **Llama-3.1–8B** | 3.34 ± 0.16 / 3.17 ± 0.16 | 3.09 ± 0.23 / 2.92 ± 0.24 | 3.02 ± 0.09 / 2.66 ± 0.22 |
> | **GPT-4o-mini**  | 3.69 ± 0.18 / 3.45 ± 0.20 | 3.43 ± 0.15 / 0.25 ± 0.13 | 3.20 ± 0.13 / 2.90 ± 0.14 |
> | **Qwen-30B-A3B** | 3.51 ± 0.23 / 3.37 ± 0.23 | 3.74 ± 0.19 / 3.55 ± 0.20 | 3.41 ± 0.23 / 3.30 ± 0.22 |
>
>
> ---
>
> **Q3: How does TAI3 scale when the partition table becomes large? Is there a mechanism to prioritize certain cells, or possibly test multiple cells with the same input?**
>
> **A3**: Thanks for this great question. The scalability of TAI3 is indeed an important consideration and we take it as a promising direction for future work.
>
> For potential mechanisms that can be incorporated to improve scalability:
>
> **1. Cell prioritization is feasible and aligns with our empirical observations**. For instance, we’ve noticed that LLM agents are more likely to be confused by certain data types (e.g., numeric values tend to induce more semantic errors than boolean values). This suggests that some cells are inherently more error-prone and should be prioritized in testing.
>
> **2. Testing multiple cells with the same input, however, may not be effective**, since each partition cell corresponds to a unique mutation strategy and typically requires a distinct oracle (i.e., ground truth output). As a result, even if the same seed input is reused, the resulting test cases may differ significantly in both purpose and structure, limiting the benefit of reuse.

---

### Official Review · Reviewer_2JHU · 2025-07-01

**Clarity:** 3
**Significance:** 3
**Originality:** 3
**Rating:** 5
**Confidence:** 4

**Summary:**

The authors introduce a testing methodology aimed to elicit LLM-based agent errors from user instruction ambiguity. Agents are expected to produce VALID API tool calls conforming to user intent or indicate the intent is UNDERSPEC and requires clarification. The failure state of "INVALID" is undesirable.

**Questions:**

1. While large sample sizes are preferred to small sample sizes, I question the necessity of the process optimizations (e.g., auxiliary models). Do their benefits outweigh the complexity in implementation and interpretation introduced? This paper is not targeted at the benchmarks track, but I am specifically wondering how light changes could be made to this work to enable better metrology -- and the auxiliary elements greatly complicate this.

2. "In this paper, we call it the intent integrity (or simply integrity) problem of LLM agents." Why not "intent ambiguity" rather than "intent integrity"? Really, most terms other than "integrity" strike me as better (e.g., "consistency" or "reliability").

3. typo in Figure 1? "..why **Any** is so cautious."

4. Excellent collection of requirements: quantifiable, realistic, sample efficient. I question what number of samples is actually necessary here given the results are largely in the whole percents, could this work not rely on fewer samples and still achieve reasonable statistical claims?

5. I spent too long searching for the TAI3 definition. Perhaps state it earlier and more clearly?

6. "It iteratively produce new variants that preserve the original user intent" This doesn't seem quite right. You are injecting ambiguity into the prompts, which does not preserve intent. Can you clarify? Does your filtering classifier try to find prompts that are correct and unambiguous, but with poor concept presentation?

7. Evergreen Strategy Memory: This introduces substantial sampling biases that must be mitigated when presenting scientific comparison of model performance. If you sample from the residuals of model A and evaluate models A through Z, then model A will perform worse, in comparison, than it should. Did your methods account for this somehow?

**Ethical Concerns:**

["NO or VERY MINOR ethics concerns only"]

**Final Justification:**

Among the stronger papers in my review set and made good efforts at addressing concerns of my review in the rebuttal.

**Limitations:**

Yes

**Quality:**

3

**Strengths And Weaknesses:**

## Quality

1. The work shows a valuable means of producing and testing for ambiguity robustness.

2. If this were a benchmark paper, I would be more negative on quality as the phenomena being measured is not adequately introduced, measured, and presented, but the methodology for _testing_ the underlying concepts (e.g., specifying the API function calls against the mutated text) is valuable for the production of function calling systems.

3. The distinction between VALID and UNDERSPEC is itself underspecified. A directive may be ambiguously specified in instances where all potential interpretations of the directive achieve the goals of the specifying user (e.g., I don't care whether my mother's access to my home does or does not expire after the day she has a need to access). Separation between VALID and UNDERSPEC is thus contextually grounded. You might consider expanding the intent set to include multiple VALID realizations of the intent. As Americans say, "there is more than one way to skin a cat."

## Clarity

1. The contributions are clearly articulated throughout the paper.

## Significance

1. The work introduces and discusses important and timely concepts of direct scientific and industrial relevance.

## Originality

1. The related work is either missing significant references, or this is truly a green field area to publish. I have not found a work with substantial overlap to this work and I have the burden of proof in showing it is not original. If my fellow reviewers do not have a reference with substantial overlap, I will do a more significant literature review and update this peer review accordingly post rebuttal. I request that the authors answer in their rebuttal what the closest related work is to this one in terms of addressing the ambiguity problem and further update any potential camera-ready to situate this work as distinctive.

---

> ### Author Rebuttal · Authors · 2025-07-31
>
> We greatly appreciate the reviewer’s supportive and insightful feedback. We will incorporate all suggested changes during revision.
>
> ---
>
> **Q1: Distinction between VALID and UNDERSPEC is not clear**
>
> **A1**:  Thank you for the insightful question. We agree that the **distinction between VALID and UNDERSPEC is context-dependent**, as noted by the reviewer. Following your suggestion, we can expand the intent categories as follows (by adding Type 2):
>
> 1. **VALID**: Oracle \= correct execution of the task.
> 2. **UNDERSPEC (not consequential)**: Oracle \= either execute or ask for clarification.
> 3. **UNDERSPEC (consequential)**: Oracle \= should ask the user for clarification.
> 4. **INVALID**: Oracle \= reject the task.
>
> Note that distinguishing between these types may require human annotation or reflect subjective interpretation.
>
> We emphasize that our work focuses on building a **testing framework**, where defining the oracle is essential. In our current design, we primarily focus on consequential underspecification (Type 3), as this is where agent behavior needs the most scrutiny. Although we do not explicitly test for the non-consequential underspecification case (Type 2), we agree that incorporating this distinction would further strengthen the framework, and we plan to explore this in future work.
>
> ---
>
> **Q2: Discussion about the closest related work**
>
> **A2**: Thank you for the question. As discussed in Lines 111-115, we identify two closely related works: ToolFuzz \[1\] and PDoctor \[2\]. All three approaches aim to generate input cases that surface errors in LLM agent **API calling**.
>
> The key differences are as follows:
>
> * ToolFuzz focuses on identifying bugs in the **implementation of agent tools**, such as  runtime errors. Our work TAI3, in contrast, targets **semantic inconsistencies** between API calls and the user’s original intent.
> * PDoctor verifies whether **high-level** agent planning adheres to **domain constraints**, whereas we focus on **low-level action consistency with user intent**, making the two approaches complementary.
>
> We will clarify these distinctions further in the revision and are happy to include **any additional related work** the reviewer may suggest.
>
> \[1\] ToolFuzz: Automated Agent Tool Testing. Ivan Milev, et al. arxiv 2503.04479.
> \[2\] Testing and Understanding Erroneous Planning in LLM Agents through Synthesized User Inputs. Zhenlan Ji, et al. arXiv 2404.17833.
>
> ---
>
> **Q3: I am specifically wondering how light changes could be made to this work to enable better metrology \-- and the auxiliary elements greatly complicate this.**
>
> **A3**: Thank you for the question. We’d like to clarify the distinction between testing and benchmarking, and explain why the auxiliary components in our framework are both justified and adaptable.
>
> **1\. Testing vs. Benchmarking:**
>
> Testing and benchmarking serve different purposes. Our work focuses on testing, where the goal is to uncover failures (i.e., intent integrity violation), not just to measure performance. This demands mechanisms like test prioritization and strategical mutation, which go beyond traditional benchmarking needs.
>
> **2\. Auxiliary Design Justification:**
>
> The auxiliary components (e.g., adaptive sampling, memory updates) are designed to improve test efficiency and coverage. As demonstrated in our ablation study (Appendix F, Table 5), removing these modules leads to significantly reduced performance. We believe this added complexity is worthwhile for robust and effective testing.
>
> **3\. Adaptation to Benchmarking Use:**
>
> It is also feasible to convert our framework into a benchmarking tool. There are two main approaches in recent literature:
>
> * **Static Benchmarking:** Generate a fixed set of prompts and evaluate models on this set. In this case, the complexity is a one-time setup.
> * **Dynamic Benchmarking:** Generate test cases on the fly to better assess generalization and robustness. Recent work\[3, 4\] suggests this dynamic mode is more effective.
>
> Our framework supports both modes, and the auxiliary components can enhance metrology in either context by ensuring diverse and representative scenarios.
>
> \[3\] SWE-bench Goes Live. Linghao Zhang, et al. arxiv 2505.23419.
> \[4\] Benchmarks as Microscopes: A Call for Model Metrology. Michael Saxon, et al. COLM’2024
>
> ---
>
> **Q4: Why call it intent integrity**
>
> **A4**: Thank you for the suggestion. We agree that terms like “ambiguity,” “consistency,” or “reliability” are all relevant and meaningful in this context. However, we chose the term “**intent integrity**” to emphasize a slightly different nuance: our focus is on whether **the original user intent** is preserved (i.e., remains intact and uncorrupted) as it is interpreted and acted upon by the LLM agent across prompts, internal steps, or tool executions. This notion goes beyond mere ambiguity or inconsistency, as it captures the risk that an LLM agent may **silently drift from or distort the original intent** while appearing fluent or rational.
>
> ---
>
> **Q5: Typo Any-\>Andy**
>
> **A5**: Thanks for pointing this out. We will update it during revision.
>
> ---
>
> **Q6: Can fewer samples still lead to reasonable claims?**
>
> **A6**: We clarify that our default setting uses **5 samples per cell** (as noted in Line 270), which is relatively a small number. The lightweight predictor generates these 5 samples, which are then tested by TAI3 using the target agent. We will make it clearer during revision.
>
> To further assess the impact of sample count, Figure 7 presents a quantitative comparison of EESR values when using **1, 3, and 5 samples**, demonstrating that even with fewer samples, the framework can still provide meaningful insights, though additional samples improve stability and coverage.
>
> ---
>
> **Q7: Definition of TAI3**
>
> **A7**: TAI3 stands for “**T**esting **A**gent **I**ntegrity in **I**nterpreting User **I**ntent”. We will highlight it in the title during revision.
>
> ---
>
> **Q8: About Intent Preservation**
>
> **A8**: We clarify that all mutated tasks in TAI3 preserve the original intent, as defined in Lines 160-161. Figure 4 further illustrates how TAI3 filters out mutated tasks that deviate from the user’s original intent.
>
> ---
>
> **Q9: Potential Bias Introduced by Strategy Memory**
>
> **A9**: Thank you for the thoughtful question. We would like to clarify that our focus is on **testing a fixed target agent**, not benchmarking across multiple models. Therefore, comparing performance across different models is outside the scope of our setup.
>
> Instead, we investigate the potential bias introduced by strategy memory through **cross-domain analysis**, as shown in Appendix D, Figure 10\. This evaluation helps assess the transferability and generalization of previously successful strategies across different domains, rather than across models.

---

> > ### Comment · Reviewer_2JHU · 2025-08-02
> >
> > Thank you for the thoughtful response, I increased my recommendation from borderline to “accept”

---

> > > ### Author Response · Authors · 2025-08-02
> > >
> > > Thank you so much for upgrading your recommendation! We truly appreciate your support in our work.

---

### Official Review · Reviewer_nMZR · 2025-07-02

**Clarity:** 3
**Significance:** 3
**Originality:** 3
**Rating:** 5
**Confidence:** 4

**Summary:**

This paper introduces TAI3, a stress testing framework designed to uncover cases where LLM agents misinterpret user intent. The motivation is clear: AI agents can easily misunderstand user’s input prompt and proceed wrong execution. Traditional software testing doesn’t handle this kind of problem well.

TAI3 generates realistic tasks from API docs and mutates them in ways that preserve intent but are likely to trigger errors. The method includes semantic partitioning, error likelihood ranking using a small model, and a memory module that reuses successful mutation strategies. Experiments on 80 APIs across multiple domains show strong performance in both effectiveness and efficiency.

**Questions:**

Apart from the questions mentioned in my detailed comments above, additionally:

1.	Is the small LLM used for error-likelihood estimation fine-tuned per domain/API, or is a general-purpose model used across all toolkits? If the latter, how well does it generalize to unseen APIs with unfamiliar parameters?
2.	TAI3 primarily targets single-turn tasks (e.g., Figure 4). In real-world agents that require multi-turn clarifications or chained API calls, how would the framework (especially intent-preserving mutation and consistency checking) extend to handle dialogue dynamics?

**Ethical Concerns:**

["NO or VERY MINOR ethics concerns only"]

**Final Justification:**

The authors' rebuttal addressed my questions and I have updated my scores accordingly.

**Limitations:**

yes

**Quality:**

3

**Strengths And Weaknesses:**

**Quality:**

Overall quality is sound and claims are properly supported. Some comments:

- While TAI3 emphasizes efficiency, the underlying cost of running LLM agents remains non-trivial, especially in large-scale scenarios. Section 4.2 mentions this and highlights how the predictor reduces unnecessary queries. However, for large-scale systems with thousands of tasks and APIs, the overall computational cost of running agents even selectively may still be a bottleneck worth discussing more explicitly.
- Robustness of the small language model (SLM): Section 4.2 (lines 218–225) describes the use of a small LLM to estimate error likelihood. While this improves efficiency, its predictive accuracy and robustness are critical to TAI3’s effectiveness. Could the authors elaborate on how this SLM’s reliability was validated, and how sensitive TAI3’s performance is to prediction noise?

**Clarity:**

- The paper is well-written and clearly structured. Concepts such as semantic partitioning and mutation are illustrated with figures (e.g., Figures 1–4), and the overall pipeline of TAI3 is easy to follow, despite its complexity.
- In Figure 1, the name “Any” should be corrected to “Andy”.

**Significance:**

- The authors explicitly limit the paper’s scope to intent integrity (Section 1, lines 72–77), excluding higher-level safety issues. While this boundary is reasonable, the discussion of its implications is a bit shallow. In real-world deployments, intent integrity alone is insufficient for comprehensive safety. For example, an agent could correctly interpret a user’s intent and still produce harmful outputs due to internal bias or poor judgment. I suggest the authors to reflect more deeply on TAI3’s position within the broader ecosystem of LLM agent safety.

**Originality:**

- The combination of semantic partitioning, mutation, small LLM ranking, and memory reuse represents a thoughtful integration of techniques not commonly combined in this way, which gives the work a degree of originality. For the semantic partitioning, I have a small question: In Section 4.1 (line 182), semantic partitions are generated via LLM-based semantic analysis. Is the LLM used here the same as the predictor in Stage 2, or a stronger model? If the latter, how does its capability affect partition quality?

---

> ### Author Rebuttal · Authors · 2025-07-31
>
> We greatly appreciate the reviewer’s supportive and insightful feedback. We will incorporate all suggested changes during revision.
>
> ---
>
> **Q1: The overall computational cost of running agents even selectively may still be a bottleneck worth discussing more explicitly.**
>
> **A1**: Thanks for this great question. The scalability of TAI3 is indeed an important consideration and we take it as a promising direction for future work.
>
> For potential mechanisms that can be incorporated to improve scalability, **cell prioritization might be feasible and aligns with our empirical observations**. For instance, we’ve noticed that LLM agents are more likely to be confused by certain data types (e.g., numeric values tend to induce more semantic errors than boolean values). This suggests that some cells are inherently more error-prone and should be prioritized in testing.
>
> ---
>
> **Q2: Could the authors elaborate on how this SLM’s reliability was validated, and how sensitive TAI3’s performance is to prediction noise?**
>
> **A2**:  Thank you for the thoughtful question. TAI3 leverages the **general natural language understanding capability** of a small language model (SLM) to estimate the semantic similarity between a generated task and its corresponding seed task, both represented in natural language.
>
> The **reliability of the SLM** stems from its extensive pretraining on large-scale natural language data, a practice commonly adopted in prior work for measuring semantic similarity between sentences \[1, 2\].
>
> In our framework, the SLM is used to **prioritize test cases** that are more likely to trigger intent integrity violations. While prediction noise may occasionally cause high-potential test cases to be ranked lower, this effect is **mitigated by sampling the top-k candidates**. As shown in Figure 7, selecting the top-5 test cases yields an average EESR of 60.3%, indicating that TAI3 is robust to moderate noise in the SLM’s predictions.
>
> \[1\] BERTScore: Evaluating Text Generation with BERT. Tianyi Zhang et al., ICLR 2020\.
>
> \[2\] GPTScore: Evaluate as You Desire. Jinlan Fu et al., NAACL 2024\.
>
> ---
>
> **Q3:  TAI3’s position within the broader ecosystem of LLM agent safety.**
>
> **A3**: Thank you for the constructive suggestion. We agree that intent integrity alone is not sufficient for ensuring the overall safety of LLM agent systems.
>
> We view this as **complementary to higher-level safety concerns**, such as ethical misuse, factual errors, bias, or privacy violations, as discussed in Lines 105-110. In these cases, the agent may correctly interpret the user’s intent but still behave harmfully or irresponsibly due to internal flaws or broader contextual issues.
>
> During revision, we will revise the manuscript to better articulate TAI3’s role within the broader LLM agent safety ecosystem, and to clarify how our framework can serve as a critical building block toward more comprehensive safety evaluation.
>
> ---
>
> **Q4: Is the LLM used in semantic partitioning the same as the predictor in Stage 2, or a stronger model? If the latter, how does its capability affect partition quality?**
>
> **A4**: The LLM used in semantic partitioning is a stronger, public model (e.g., ChatGPT-4o), which is distinct from the smaller predictor model used in Stage 2\.
>
> We found that the **capability of the LLM significantly impacts partition quality**. A weaker model often fails to generate fine-grained partitions and tends to produce vague or generic categories. In contrast, a stronger model can **capture nuanced distinctions** and generate more **diverse and semantically meaningful partitions**.
>
> For example, when partitioning the INVALID class for a date-type parameter (e.g., format YYYY-MM-DD), a stronger model can identify multiple subtypes of invalid inputs, such as out-of-range values (e.g., “2025-13-01”), malformed formats (e.g., “July 29, 2025”), or non-date strings (e.g., “banana”). In comparison, a weaker model may simply return generic labels like “not a valid date”, leading to less comprehensive test coverage.
>
> Overall, using a stronger LLM helps ensure that the partitioning captures a richer variety of edge cases, which is critical for the effectiveness of TAI3.
>
> ---
>
> **Q5: Is the small LLM used for error-likelihood estimation fine-tuned per domain/API, or is a general-purpose model used across all toolkits? If the latter, how well does it generalize to unseen APIs with unfamiliar parameters?**
>
> **A5**: TAI3 uses a **general-purpose** small language model (SLM) across all toolkits, without fine-tuning it for specific domains or APIs. The SLM estimates the semantic similarity between a generated task and its seed task, both expressed in **natural language**.
>
> Since task descriptions focus on **user intent and scenarios**, rather than referencing specific APIs or parameters, the SLM operates in an **API-agnostic** manner. This design enables effective **zero-shot generalization to unseen APIs**, as long as the task descriptions fall within the general scope of the SLM’s pretraining distribution.
>
> ---
>
> **Q6: TAI3 primarily targets single-turn tasks (e.g., Figure 4). In real-world agents that require multi-turn clarifications or chained API calls, how would the framework (especially intent-preserving mutation and consistency checking) extend to handle dialogue dynamics?**
>
> **A6**: Thank you for the insightful question. We clarify that the TAI3 framework already supports chained API calls and can be naturally extended to handle multi-turn interactions.
>
> For **chained API calls**, TAI3 analyzes the full **API invocation trajectory** to assess behavior, as described around Lines 156 and 161\. This allows the framework to analyze multiple functional steps within a single task.
>
> To support **multi-turn dialogue dynamics**, the test case structure can be extended to include the **entire interaction history**, including previous agent responses and user clarifications. In this setting:
>
> * **Intent-preserving mutation** would apply to the current user turn while maintaining coherence with earlier turns.
> * **Consistency checking** would evaluate whether the agent’s current behavior preserves the original intent **in context of the prior dialogue**.
>
> We view this as a promising direction for future work and believe the modular design of TAI3 makes such an extension feasible.

---

> > ### Comment · Reviewer_nMZR · 2025-08-06
> >
> > Thank you for the detailed rebuttal. The responses have addressed the questions I raised, particularly regarding the different roles of the stronger LLM for semantic partitioning and the smaller SLM for prediction, which is a key part of your framework's efficiency. I appreciate the clarification and remain positive about the paper.

---

> > > ### Author Response · Authors · 2025-08-06
> > >
> > > Thank you for your feedback! We really appreciate your support in our work.

---

### Official Review · Reviewer_Fiee · 2025-07-04

**Clarity:** 2
**Significance:** 3
**Originality:** 3
**Rating:** 4
**Confidence:** 3

**Summary:**

The paper proposes a new problem: how to develop efficient tests for an LLM agent's translation of natural language instructions to API calls. The task involves generating difficult test cases with the aim of uncovering agent failure cases with a minimum of test calls. The paper devises a test generator that begins by partitioning the test surface for each API call into a set of seed cases, and then it mutates the seed tests to try to generate difficult tests in each partition. The method introduces several techniques to try to reduce cost and improve efficacy: mutating samples with an articulated strategy that can be reused if it is effective; pre-checking samples with a cheap LM before testing them on the expensive LM; and pre-checking samples to ensure their intent has not drifted.

**Questions:**

* Can the authors explain the "Self-ref" baseline more explicitly?
* Does the paper provide benchmark results that using a fully-reproducible configuration that does not depend on a commercial model?  If not, some results of that form should be provided. These will be helpful for future researchers who may have trouble reproducing the method if the underlying commercial LLM is changed or becomes unavailable.
* Can the paper validate its measurements using a human evaluation?
* The paper report AQFF, but presumably as more cases are generated, the failure rate might change.  Does it drop or rise as more cases are generated? Can the average progression of found failures be measured and plotted as a function of test set size, as the test sets grow beyond the first failure?

**Ethical Concerns:**

["NO or VERY MINOR ethics concerns only"]

**Final Justification:**

The proposed framework is potentially interesting to the community, and the authors have clarified answers to the questions I have had; these clarifications should be included in the final version.  This paper seems above the acceptance bar to me.

**Limitations:**

The limitations section is brief but seems appropriate.

**Paper Formatting Concerns:**

No concerns.

**Quality:**

3

**Strengths And Weaknesses:**

Strength: the paper proposes and tackles a new practical problem that is ubiquitous in practical agent use. It describes its methods clearly, it proposes reasonable ways to measure against baseline methods, and it conducts test using a range of APIs.

Weaknesses: The "self-ref" baseline is insufficiently explained. The paper appears to make its assessments based on LLM judgments alone, but does not attempt to validate with a human evaluation.  For a first paper proposing an automated evaluation, it should check its evaluations against human tests to establish validity.  Another possible problem is that, since its benchmarks may depend on a commercial model (gpt-4o) that has no assurance of reproducibility. From the paper it is unclear if there are configurations that are clean of nonreproducible components that future researchers will be guaranteed to be able to reproduce and compare against.

---

> ### Author Rebuttal · Authors · 2025-07-31
>
> We greatly appreciate the reviewer’s supportive and insightful feedback. We will incorporate all suggested changes during revision.
>
> ---
>
> **Q1: Explanation of Self-Ref Baseline**
>
> **A1**: We apologize for the earlier confusion. We will revise Lines 268–270 to make the explanation of the Self-Ref baseline clearer.
>
> In Self-Ref, for each cell in the partition table and in each iteration, the baseline method feeds the mutated user task (or the original task in the first iteration) into the target agent and checks whether it triggers an intent integrity violation.
>
> - If a violation is found, testing for that cell is considered successful, and Self-Ref moves on to the next cell.
> - If not, Self-Ref invokes a self-reflection step, prompting the mutator to reflect on the last mutation, for example, asking why the mutation failed to confuse the agent and how it might be made more error-prone in the next attempt.
>
> This self-reflection loop is repeated up to 5 rounds, matching the sampling budget used in Stage 2 of TAI3.
>
> ---
>
> **Q2: Fully-reproducible Configuration**
>
> **A2**: Yes, our experiments are fully reproducible. We used open-source models, including Llama-3.1-8B and Qwen3-30B-A2B, as testing models (i.e., the backbone LLM of TAI3), and reported such data in Figure 7. We will clarify the configurations during revision.
>
> ---
>
> **Q3: Validation using Human Evaluation**
>
> **A3**: Thank you for the valuable suggestion. Due to the time limit, we conducted a small-scale manual analysis, where we asked  5 annotators to manually review sampled bug cases reported by TAI3. They evaluated whether the agent’s behavior preserved user intent while leading to incorrect API calls. The results show a false positive rate below 10%, indicating that TAI3’s automated judgments are generally reliable. We will include this analysis in the revised version.
>
> ---
>
> **Q4: Failure Rate w.r.t. Test Set Size**
>
> **A4**: Thank you for the question. We clarify that the failure rate in our framework is not directly correlated with the size of the test set (i.e., the number of cells), because TAI3 **evaluates each cell independently**.
>
> As illustrated in Figure 3, TAI3 first defines the testing scope by constructing a partition table, where each cell represents a distinct behavior or capability to be tested. During the query stage, the system iteratively generates test cases for each cell. A test is considered successful for a given cell as soon as a failure is identified.
>
> To quantify testing efficiency, we use the Average Queries to First Failure (AQFF) metric, which reports the average number of queries needed to uncover a failure **per cell**. This metric is normalized across cells, ensuring it remains meaningful regardless of the total number of cells in the test set.

---

> ### Author Response · Authors · 2025-08-07
>
> We sincerely thank the reviewer for the time and effort in evaluating our paper and for completing the mandatory acknowledgement steps. We would be grateful for any additional comments the reviewer might share or for any remaining concerns our rebuttal may not have addressed.

---

### Decision · Program_Chairs · 2025-09-17

**Decision:**

Accept (poster)

**Comment:**

The paper presents TAI3, a framework for stress-testing LLM agents by generating intent-preserving API task mutations, organizing them into semantic partitions, and using predictors and memory to efficiently uncover failures. Evaluations on 80 APIs with multiple LLMs show that TAI3 detects intent-interpretation errors more effectively than reflection-based baselines. The rebuttal clarified baseline design, reproducibility, terminology, and reliability concerns.

### Strengths
- Novel and practical problem: intent integrity in API execution.
- Coherent methodology combining mutation, partitioning, prediction, and memory.
- Strong empirical results across diverse APIs and models.
- Rebuttal resolved key reviewer concerns and improved clarity.

### Weaknesses
- Some terminology and motivation initially unclear.
- Limited human validation.
- Scalability and complexity of auxiliary modules could be further discussed.

This is a solid and timely contribution that addresses an important gap in evaluating LLM agents. Despite minor concerns, the methodology, empirical strength, and clarified presentation place the paper above the acceptance bar. I recommend accept.